# Myalgic Encephalomyelitis/Chronic Fatigue Syndrome: Impact on Quality of Life (QoL) of Persons with ME/CFS

**DOI:** 10.3390/medicina60081215

**Published:** 2024-07-27

**Authors:** Nina L. Muirhead, Jui Vyas, Rachel Ephgrave, Ravinder Singh, Andrew Y. Finlay

**Affiliations:** 1Department of Dermatology, Buckinghamshire Healthcare NHS Trust, Amersham HP7 0JD, UK; nina.muirhead@btinternet.com; 2Centre for Medical Education, School of Medicine, Cardiff University, Cardiff CF14 4YS, UK; 3Patient Research Partner, Gloucestershire, UK; 4Medical Research Council, Polaris House, Swindon SN2 IFL, UK; ravinder.singh@mrc.ukri.org; 5Division of Infection and Immunity, School of Medicine, Cardiff University, Cardiff CF14 4YS, UK; finlayay@cardiff.ac.uk

**Keywords:** myalgic encephalomyelitis, chronic fatigue syndrome, quality of life

## Abstract

*Background and Objectives*: We previously reported on the impact of myalgic encephalomyelitis/chronic fatigue syndrome (ME/CFS) on the QoL of persons with ME/CFS and their family members. Here, we present the findings of the impact on the QoL of individuals with ME/CFS whose family members did not participate in the survey. *Materials and Methods*: A prospective multinational online survey was disseminated via patient charities, support groups and social media. Persons with ME/CFS completed the EuroQoL questionnaire (EQ-5D-3L). *Results*: Data were analysed from 876 participants from 26 countries who reported a health care professional diagnosis of ME/CFS. In total, 742 participants identified as female, 124 male and 10 preferred not to say. The mean age of the participants was 47 years (range 18–82), and the mean time to diagnosis was 14 years. The mean overall health status on a visual analogue scale for people with ME/CFS was 36.4 (100 = best health). People with ME/CFS were most often affected by inability to perform usual activities (n = 852, 97%), followed by pain (n = 809, 92%), impaired mobility (n = 724, 83%), difficulty in self-care (n = 561, 64%) and least often affected by anxiety and depression (n = 540, 62%). *Conclusions*: The QoL of people with ME/CFS is significantly affected globally. There was no significant difference in quality of life compared with previously published data on those with ME/CFS who did have a family member complete the family member quality of life questionnaire (FROM16). Contrary to popular misconception, anxiety and depression are the least often affected areas in persons with ME/CFS who are most impacted by their inability to perform usual activities.

## 1. Introduction

Myalgic encephalomyelitis/chronic fatigue syndrome (ME/CFS) is a chronic, complex, and debilitating disease affecting an estimated 20 million people worldwide. It is characterised by the worsening of symptoms after physical, mental or emotional exertion, known as post exertional malaise (PEM). PEM is often delayed in onset by hours or days, disproportionate to the exertion and can last for hours, days, weeks or longer [1]. Other diagnostic features are disabling fatigue, unrefreshing sleep and cognitive impairment. Multisystem symptoms include cardiovascular, respiratory, neurological, musculoskeletal, abdominal, flu-like symptoms, sore throats, headaches, widespread pain, tinnitus and orthostatic intolerance; people with ME/CFS are often unable to work full time, with many being housebound or bedbound [1,2,3].

In our previous international study [4], people with (ME/CFS) and their partners/family members reported on the impact on their quality of life (QoL) using validated questionnaires: EQ-5D-3L [5] for people with ME/CFS and FROM-16 [6] for their family members. However, many participants with ME/CFS only submitted data about their own QoL and either did not provide family member information or their family members did not complete the FROM-16. ME/CFS often results in isolation [7] and may result in reduced contact with family members. It is recognised that the trauma of ME/CFS stigma and lack of social support [8] can cause family members to become suspicious that their relative may be malingering or mentally ill [9], adding further burden to the ME/CFS patient. The aims of this paper were to publish the QoL data for those patients who did complete the EQ-5D-3L but did not have a partner/family member complete the FROM-16 survey. The burden on QoL of 876 patients from 26 different countries is presented here.

## 2. Methods

The data for this study were collected from a multinational cross-sectional survey [4] to assess the impact of ME/CFS on the quality of life of people with ME/CFS and their family members using the EQ-5D-3L Dimension (EQ-5D) [5] and FROM-16 [6]. This study was co-designed by patients and clinical researchers. A survey was distributed via social media platforms and ME/CFS organisations and websites; the data were collected and managed using a secure Research Electronic Data Capture web platform (REDCap) hosted at Cardiff University, United Kingdom [10,11]. For the purpose of this paper, only data from participants whose family members did not complete the FROM survey were analysed. For more information on the original study design, please refer to the publication [4].

Ethical approval was granted by the Cardiff University School of Medicine ethics committee (11 September 2020) (SMREC 20/86). An amendment to publish data of participants with ME/CFS whose family members did not respond to the survey was approved on 18 January 2024. All participants gave informed consent to participate in the study before taking part. Participants completed a demographic questionnaire and the EQ-5D-3L questionnaire. The EQ-5D-3L is a standardised validated instrument used to assess an individual’s health status in the five dimensions of mobility, usual activities, self-care, pain and discomfort and anxiety and depression. The questions offer three possible responses for each dimension; mobility: no problems, some problems walking about or confined to bed; usual activities: no problems, some problems or unable to perform usual activities; self-care: no problems, some problems washing or dressing or unable to wash and dress; pain and discomfort: no pain, moderate pain or extreme pain; and finally, anxiety and depression: none, moderate or extreme. Each response is assigned a value from 1 to 3, creating a five-digit code that represents the individual’s EQ-5D self-reported health state’ or ‘EQ-5D profile’. There are 234 possible health states within the EQ-5D-3L system. These profiles can be translated into a single score known as the ‘EQ-5D index value’, where ‘1’ denotes perfect health and ‘0’ represents death. Scores less than 0 signify a health state considered worse than death. Additionally, overall health status is captured on a visual analogue scale (VAS) ranging from 0 (worst imaginable health) to 100 (best imaginable health).

### Statistical Analysis

Only data from participants with a reported diagnosis of ME/CFS from a health care professional was included in the final analysis. Duplicate entries were identified by email addresses and matching demographics. Only the second was analysed. Microsoft Excel (version 2406) and SPSS (version 29.0) were used for data handling and statistical analysis. The Mann–Whitney U test and Spearman Rank coefficient were used for statistical correlation of non-parametric data.

## 3. Results

The survey was conducted from 1 December 2020 to 31 March 2021. It was started 2980 times, but one participant withdrew consent, so 2979 records were generated. After the exclusion of duplicates, 1479 records were fully completed by both patients and family members/partners, and 1418 of these were included in the previously published final analysis. Overall, 1500 participants with ME/CFS started the EQ-5D, 1259 people with ME/CFS completed the EQ-5D survey, and 66 of these participants did not complete the global questionnaire of the EQ-5D (VAS) and were excluded from the analysis. Further exclusions included 283 participants whose family members had consented to the study but had not completed the study and 32 participants who did not have a formal diagnosis. Two were under the age of 18 years and were also excluded from the analysis (see Figure 1). The data reported in this study are based on the 876 people with ME/CFS who completed the demographic and QoL survey and who did not have family members/partners complete the FROM-16.

### 3.1. Demographic Profile of Participants

Table 1 shows the demographic characteristics of the study participants included in this latest analysis. While individuals with ME/CFS from around the globe participated, the majority were from the United Kingdom (59.2%) (Table 2). The mean duration since diagnosis was 14.6 years (median 11 years; range 1–50), with 14 participants diagnosed for <1 year and 7 participants > 50 years. Consistent with the ME/CFS demographic, the majority of the respondents were female (87.6%, n = 742), with 1.2% (n = 10) opting not to disclose their gender. From the sample, 277 participants (31.6%) lived alone and 26.3% (n = 230) reported having another chronic condition.

### 3.2. EQ-5D Health Profiles

Of the possible 243 EQ-5D profiles, participants exhibited 99 distinct profiles. The most frequently observed EQ-5D profile was 22322 (8.2%, n = 72), indicative of some problems with mobility and self-care, inability to perform usual activities and moderate pain/discomfort and anxiety/depression. This was closely followed by the profile 22222 (7.5%, n = 66), representing moderate problems in all dimensions. Notably, only two participants reported a profile of 11111, indicating no problems in any dimension, and eight participants reported a profile of 33333, indicating extreme problems in all dimensions. The 10 most common EQ-5D profiles accounted for 53.5% of the total EQ-5D-3L profiles, as detailed in Table 3. As mentioned above, the EQ-5D profile can be converted into a summary score (index value), allowing for comparison with the general population. The EQ-5D index values ranged from −0.073 to 1, with a mean of 0.378 (median 0.31; range −0.073–1). Two participants reported an EQ-5D index value of 1, indicating perfect health, while 21 participants (2.4.%) reported a negative value, indicating a state worse than death.

The dimension most affected was usual activities, with 97.3% (n = 852) of participants reporting either an inability to perform their usual activities (47.5%, n = 416) or experiencing some problems (49.8%, n = 436). Pain/discomfort followed as the next most affected dimension (92.3%, n = 809), with 26.3% (n = 231) of the participants experiencing extreme pain and discomfort. Mobility issues were prevalent, with 82.6% of patients experiencing problems with mobility, with 11.5% (n = 101) confined to bed and 71.1% (n = 623) experiencing some problems with walking. With regard to self-care, 65.2% (n = 512) had some problems with self-care and 5.6% (n = 49) were unable to wash and dress themselves. Anxiety/depression was the least affected dimension (61.6%), with 13.9% (n = 122) experiencing extreme and 47.7% (n = 418) experiencing moderate anxiety/depression, while 38.4% (n = 336) did not experience any anxiety or depression. The VAS index values were not normally distributed, but mean values were calculated to enable comparison with existing published data. The mean EQ-5D VAS score of patients with ME/CFS was 36.4 (median 31.5, interquartile range 22–50, range 0–100). A statistically significant moderate positive correlation (correlation coefficient 0.49, *p* < 0.001) was found between the reported VAS score and EQ-5D index value. The mean EQ-5D VAS score for people who lived alone was 33.5 (median 30, interquartile range 20–43, range 0–85). In contrast, the mean EQ-5D VAS score for participants who did not live alone was 37.8 (median 33, interquartile range 24–50, range 0–100). A statistically significant difference in VAS scores was observed between ME/CFS patients who did and did not live alone (*p* = 0.001, Mann–Whitney U test). Furthermore, Spearman’s Rank correlation demonstrated a statistically significant weak negative correlation (correlation coefficient −0.109, *p* = 0.001), indicating that participants with ME/CFS who lived alone had a lower QoL. No statistical difference was found between the EQ-5D index values of those who lived alone and those who did not live alone using the Mann–Whitney U test (*p* = 0.177).

## 4. Discussion

The most common EQ-5D-3L profiles reveal that individuals with ME/CFS encounter problems in all domains; their problems are not limited to a single dimension. Only one of the top ten most common profiles in this study reported a level 3 for anxiety and depression. In addition, anxiety and depression was the least affected domain, in keeping with our previous study [4].

A higher EQ VAS score demonstrates a better QoL. In this study, the mean EQ VAS score was 36.4, and the mean EQ VAS score for the UK population is 82.75 [12], demonstrating the huge detrimental impact of ME/CFS on the QoL of individuals with this condition. Similarly, the EQ-5D index value in this study was found to be 0.38, whereas the EQ-5D index value for the UK population norm is 0.86 [12], emphasising the negative impact of ME/CFS on QoL. Comparing the data with that of those study participants who had provided partner/family information may provide further insights into the understanding of the influence of ME/CFS on isolation and altered family relationships.

Of those participants from our previous analysis who did have a partner or family member complete the FROM-16 questionnaire, 11.1% lived alone; in this study, of those who did not have a family member or partner complete FROM-16, 31.6% lived alone. Interestingly both the mean EQ-5D Index value and the mean VAS score were higher in this participant cohort as opposed to our previous analysis. As reported above, the mean EQ-5D index value in this study was 0.38, and the mean VAS score was 36.4. In contrast, in patients whose family members responded, the EQ-5D index value was 0.36, and the mean VAS score was 33.8 (0 = worst, 100 = best). However, this study demonstrated that the people with ME/CFS who lived alone had a statistically significant decrease in their VAS score, indicating a poorer QoL compared to those who did not live alone. This highlights the issue of isolation in chronic disease, which has implications for social care. In other research, social isolation has been identified as an independent risk factor for type 2 diabetes, cardiovascular disease and all-cause mortality, which suggests that the impact of social isolation on the disease burden of ME/CFS would be important to investigate further [13].

From the data collected, there was a significant overlap in the ten most frequent EQ-5D health states of participants from the two groups analysed, with seven out of the top ten health states being the same for both groups (those with and without family members/partners completing the FROM-16). The most frequent profile experienced by participants who did not have a family member complete the FROM-16 questionnaire was 22322 (8.2%, n = 72), indicating moderate problems in all dimensions but inability to perform usual activities. This was very similar to the most frequent profile in our previous study (22321); however, it also demonstrated that patients whose family members did not respond to the survey had more of an impact in the dimension of anxiety and depression. Interestingly, the frequency of impact was similar in both studies, with usual activities affected the most, followed by pain/discomfort, mobility, self-care, and anxiety/depression. However, whilst none of the 10 most frequent profiles in the previous study demonstrated a level ‘3’ (extreme) in the dimension of anxiety and depression, the 10th most frequent EQ-5D profile in our study demonstrated extreme anxiety and depression. It is possible that people with ME/CFS who did not have a family member/partner complete the FROM-16 are similarly impaired by ME/CFS in the domains of usual activity and pain/discomfort but may experience increased anxiety and depression as they lack family support.

The strengths of this study were the large international reach and large number of respondents. One of the strengths of this paper is the ability to directly compare the 876 people with ME/CFS who did not have a family member respond to the survey with the data previously published on the 1418 people with ME/CFS who did. A strength of the style of the questionnaire was that it was short and had been trialled with patients, enabling the participation of patients who are more severely affected and often too disabled to travel to or attend an outpatient clinic. Patients and clinical researchers worked as equal partners from the outset of the research and were involved in the co-design of the research and application for ethics.

Limitations of the study include open participation recruitment bias towards English-speaking self-selected people active on social media. Additionally, data on ethnicity were not captured. The EQ-5D has multiple language versions tailored for different countries with different EQ-5D value sets. Since the UK version of the EQ-5D questionnaire was used for the study and the majority of our participants were from the UK, we calculated values for all participants using the UK value set. Information was not collected on the reasons for non-response by partners and family members; however, many individuals with ME/CFS who were participating in the study contacted the research team directly to explain their illness had resulted in isolation and lack of support. The VAS score was not collected on all participants who dropped out; therefore, it is not possible for us to comment on the possibility that those who were missed were generally in the very severe category. ME/CFS often results in isolation, [7] prejudice, disbelief and stigma [7,14] related to family, friends, health and social care professionals, and teachers’ lack of understanding of the disease.

The information presented here is increasingly relevant in the context of the global impact of COVID-19. ME/CFS is thought to affect a subset of patients with Long COVID, which will result in an increased economic and social burden, including the impact on QoL. The negative impact of Long COVID on family members’ QoL was evident early in the pandemic [15]. Subsequent research on a small cohort of people with ME/CFS, Long COVID and healthy controls showed similar self-reported impact of ME/CFS and Long COVID on most of the EQ-5D domains, including usual activities and significantly worse impact than for healthy controls [16]. UK research using validated QoL questionnaires including EQ-5D-5L in Long COVID clinics showed striking levels of functional impairment and low QoL. This impairment is mainly driven by fatigue, causing a significant impact on the ability to work and care for others [17].

In recent research on health scores for ME/CFS, it has been noted that fatigue and cognitive impairment also have a significant impact and that these are not represented in the EQ-5D domains [18]; this would be an important future area to explore to advance our understanding of how QoL may be improved for ME/CFS patients worldwide.

## 5. Conclusions

These data from a large-scale international survey representing 26 countries demonstrate the worldwide impact of ME/CFS on quality of life using the EQ-5D-3L.

## Figures and Tables

**Figure 1 medicina-60-01215-f001:**
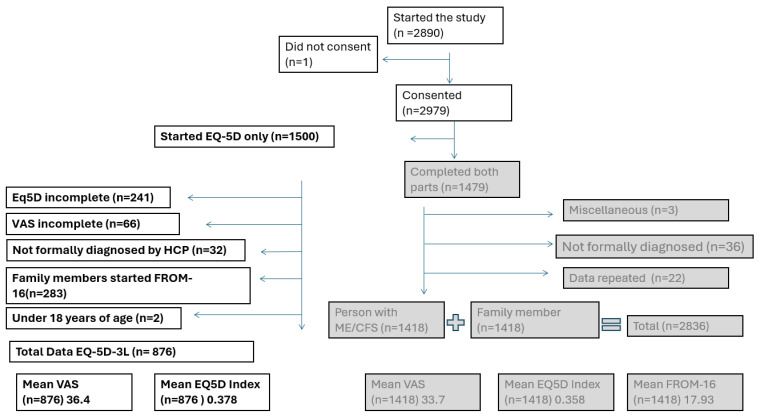
Participant numbers. Flow diagram demonstrating the basis for participant inclusion/exclusion from the analysis of the study and previously published data (greyscale). New data shown in bold show the remaining 876 people with ME/CFS identified for analysis. EQ-5D, Euroquol 5 Dimensions; ME/CFS: myalgic encephalomyelitis/chronic fatigue syndrome; VAS, visual analogue scale.

**Table 1 medicina-60-01215-t001:** Participant demographic characteristics. ME/CFS: myalgic encephalomyelitis/chronic fatigue syndrome.

	Person with ME/CFS
Number	876
Years since diagnosis (median; range)	14.6 (11; 1–50)
Age (median; range)	46.7 (47; 18–82)
Female	742 (84.7%)
Male	124 (14.2%)
Other	10 (<1%)
Lives alone	277 (31.6%)
Additional chronic condition	230 (26.3%)

**Table 2 medicina-60-01215-t002:** Countries and participants.

Country	Number of Responses People with ME/CFS	Percentage
United Kingdom	519	59.25
United States of America	80	9.13
Australia	63	7.19
Canada	29	2.63
Sweden	27	3.08
Norway	26	2.97
Germany	25	2.85
New Zealand	23	2.63
Italy	14	1.60
Ireland	13	1.48
Netherlands	11	1.26
Spain	8	0.91
France	4	0.46
South Africa	4	0.46
Switzerland	4	0.46
Belgium	4	0.46
Denmark	4	0.46
Japan	3	0.34
Finland	3	0.34
Hungary	3	0.34
Iceland	2	0.23
Argentina	2	0.23
Austria	2	0.23
Malaysia	1	0.11
Uruguay	1	0.11
Brazil	1	0.11

**Table 3 medicina-60-01215-t003:** The ten most frequent EQ-5D health states of participants with myalgic encephalomyelitis/chronic fatigue syndrome, sorted according to EQ-5D value severity.

EQ-5D	EQ-5D State	EQ-5D Value	Frequency	% Frequency
Least severe	11221	0.73	33	3.77
	11222	0.667	39	4.45
	21221	0.659	37	4.22
	21222	0.596	50	5.71
	22221	0.566	47	5.36
	22222	0.503	66	7.53
	22321	0.301	55	6.28
	22322	0.238	72	8.22
	22332	0.151	44	5.02
Most severe	22333	0.09	26	2.97

## Data Availability

Data are available upon reasonable request. The authors agree to share data on reasonable request.

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
