# Peer review of "Myalgic Encephalomyelitis/Chronic Fatigue Syndrome: Impact on Quality of Life (QoL) of Persons with ME/CFS"

_medicina, 2024, doi:10.3390/medicina60081215_

Round 1

Reviewer 1 Report

Comments and Suggestions for Authors

This is a very helpful paper in giving a good idea of the gradations of severity across the spectrum of people with the conditions. Although simply an addition of the data from those who did not have a family member who also filled in the questionnaires, it helps.

1-    It corroborates the earlier data

2-    It argues against catastrophizing being a predominant problem in the patient group (which would cause dramatically worse responses than by family members)

3-    it gives an idea of how many people are so isolated with the disease, or treated as if they had a non-real illness, that they had no family member available or willing to also fill in their questionnaire.

Overall, this and your previous paper add significantly to the understanding of symptoms and severity in these conditions.

In terms of the generalizability of the data, I think your approach was strong. One could argue that selecting out support group members would get the sicker group of people with the conditions. But one can easily make the point that the 20% in the new group that did not complete the questionnaires suggest that these had the worst brain fog and disability, and therefore did not complete the questionnaires. Leaving the most disabled 20% “underrepresented” in the final data.

I think this is a modest concern, but I see no good way to have gotten around that. Except perhaps in future surveys to simply have people give a VAS score of overall disability when they first enroll. And then seeing how these correspond to the dropouts when doing the future participant flow diagrams.  

I think the approach of doing the research by mail in this often too disabled to come to the clinic population is excellent. Especially as there are no expensive medications, resulting in little funding for these conditions.

Especially given as there are no necessary lab markers, and follow-up and assessment is subjective. Which now becomes a benefit for researching these conditions to counteract some of the difficulties, allowing the studies to be done at lower cost and without people needing to travel to the clinic.

Overall, well done!

Overall, an excellent study that will help clarify the degree of disability from these conditions.

Recommended for publication with no changes needed

Author Response

Dear Reviewer,

Thank you and for acknowledging the following strengths:

1-    It corroborates the earlier data

2-    It argues against catastrophizing

3-    it gives an idea of how many people are so isolated with the disease

It is possible that the 20% that did not complete the questionnaires suggest that these had the worst brain fog and disability, and therefore did not complete the questionnaires, but the data is not available to support this. Collecting a VAS score of overall disability when patients first enrol could identify possible reasons for dropout and this was identified as a potential future point of interest by Reviewer 1, thank you for this suggestion.

Thank you for highlighting the strength of the methodology in reaching the more severe you made this point in several ways:

  • often too disabled to come to the clinic population
  • travel to the clinic
  • strong approach for generalisability of the data in relation to severity

Thank you for stating that this adds significantly to the understanding of symptoms and severity in these conditions.

In response to your comments, we have added a strength of the study in the discussion (Page 6, Paragraph 5  Line 213-215:

A strength of the style of questionnaire was that it was short and had been trialled with patients, enabling the participation of patients who are more severely affected and often too disabled to travel to or attend an outpatient clinic.

We have also added a limitation on Page 7 Paragraph 1  Line 227-229:

The VAS score was not collected on all participants who dropped out, so it is not possible for us to comment on the possibility that those that were missed were generally in the very severe category.

Reviewer 2 Report

Comments and Suggestions for Authors

Dear Authors,

Your research is very interesting. It reveals important conclusions about persons with ME/CFS who do not have family support.

I have several comments:

Abstract: The abstract reflects the article and the key words are relevant.

Methods: The methods are described well. However please add an explanation which country scoring did you use to calculate EQ-5D score. Probably it was the UK scoring. I would advise to add a limitation in the Discussion section that UK scoring is applied for international participants.

Statistical Analysis: Please add the version of SPSS you used. Please add as well the descriptive statistics. Some of your results are presented as mean; other as median, and sometimes you reported the range and SD. The correct way to present numerical variables is by using both median and range (or median and interquartile range; for normally distributed data mean and sd). I understand that you need the mean in order to compare your results with other studies. So you can add a brief explanation why the mean that appears within the text. The SD is not necessary to be reported since the variables are non-parametric.

Results: All medians should be accompanied with the range so please redact the text.

Table 1: please add the % as well. For the numerical variables a short information in parentheses should be added: (median; range).

Table 2: please add the % as well.

Sometimes EQ-5D is written as EQ5D so please add the dash.

Please note that the average could be any of the following: the mean, the median, the mode so please avoid using it. The measure of the central tendency should be specified.

Table 3: for EQ-5D scores use 3 decimals, and for % use 2 decimals. Some are omitted.

Discussion: Please capitalize the entire word COVID and use it as COVID-19 or SARS-CoV-2. Add the limitation mentioned above as well.

I want to add an advice how to deal with international participants. As you know EQ-5D is officially translated to many languages so it would be easy to start the survey with the language question and depending on the answer to show the questionnaire in the desired language.

Author Response

Dear Reviewer,

  1. Methods: The methods are described well. However please add an explanation which country scoring did you use to calculate EQ-5D score. Probably it was the UK scoring. I would advise to add a limitation in the Discussion section that UK scoring is applied for international participants.

 Thank you for this comment we have now added in this statement in the discussion Page 6 Paragraph 6 Line 220-224:

The EQ-5D has multiple language versions tailored for different countries with different EQ-5D value sets. Since the UK version of the EQ-5D questionnaire was used for the study and the majority of our participants were from the UK we calculated values for all participants using the UK value set

  1. Statistical Analysis: Please add the version of SPSS you used.

Thank you for the comment the version number for SPSS has been added (Microsoft Excel and SPSS (version 29.0) were used for data handling and statistical analysis Page 2 Paragraph 4 Line 90)

Please add as well the descriptive statistics. Some of your results are presented as mean; other as median, and sometimes you reported the range and SD. The correct way to present numerical variables is by using both median and range (or median and interquartile range; for normally distributed data mean and sd). I understand that you need the mean in order to compare your results with other studies. So you can add a brief explanation why the mean that appears within the text.

Thank you for your comment we have added the following statement (Page 5 Paragraph 2 Line 148-149)

The VAS index values were not normally distributed but mean values were calculated to enable comparison with existing published data.

The SD is not necessary to be reported since the variables are non-parametric.

The SD has been removed (page 4 paragraph 1 line 135, and page 5 paragraph 2 lines 150,154 and 155), an interquartile range has been provided for the VAS index scores (Page 5 Paragraph 2 Lines 151,154,155).

  1. Results: All medians should be accompanied with the range so please redact the text. All medians are associated with the ranges. We have also added the range to Page 3 Paragraph 2 line 116 where it was inadvertently omitted.

The mean duration since diagnosis was 14.6 years (median 11 years; range 1-50)

  1. Table 1: please add the % as well. For the numerical variables a short information in parentheses should be added: (median; range).

This has been added in table 1 for the relevant numerical values.

  1. Table 2: please add the % as well.

A column with the percentages has been added to table 2

  1. Sometimes EQ-5D is written as EQ5D so please add the dash.

All instances of EQ5D have been replaced with EQ-5D. 8 substitutions have been made (Page 4 paragraph 1 Line 132 and 134, Page 5 paragraph 1 Line 136, Page 5 Paragraph 2 Line 152, 153, 154, 161, Page 6 Paragraph 3 Line 182)

  1. Please note that the average could be any of the following: the mean, the median, the mode so please avoid using it. The measure of the central tendency should be specified.

Thank you for this comment. The following substitutions have been made:

The mean EQ-5D VAS score of patients with ME/CFS was 36.4 (SD 19.2, median 31.5, range 0-100). (Page 5 Paragraph 2 Line 150)

  1. Table 3: for EQ-5D scores use 3 decimals, and for % use 2 decimals. Some are omitted.

Thank you for your comment. Some EQ-5D index values only have 2 decimal points, therefore these have not been changed. However, we did notice an error for 3 of the EQ-5D index values and these have been changed in the table.

Two decimals have been inserted for all the percentage values in Table 3.

  1. Discussion: Please capitalize the entire word COVID and use it as COVID-19 or SARS-CoV-2. Add the limitation mentioned above as well.

Thank you for this comment.  All instances of COVID have been capitalized (Page 7, paragraph 2 Lines 234, 236, 238 and 239). Where COVID is mentioned on its own this have been replaced by COVID 19. For Long COVID only the word COVID has been capitalized.

The limitation of using the UK EQ-5D index values has been added as in this statement in the discussion

The EQ-5D has multiple language versions tailored for different countries with different EQ-5D value sets. Since the UK version of the EQ-5D questionnaire was used for the study and the majority of our participants were from the UK we calculated values for all participants using the UK value set. (Page 6 Paragraph 5 Line 221-223 and Page 7 Paragraph 1 Line 224)

We have also added in another limitation: Additionally, data on ethnicity was not captured (Page 6 Paragraph 5 line 220 )

  1. I want to add an advice how to deal with international participants. As you know EQ-5D is officially translated to many languages so it would be easy to start the survey with the language question and depending on the answer to show the questionnaire in the desired language.

Thank you for this comment. We agree entirely with this comment and for future studies we will endeavor to make this change.
